# Explosion-Suppression Characteristics of Nonmetallic Spherical Spacers on Propane-Air Mixtures in Confined Space

**Yangyang Yu [1,2], Lehai Liu [1], Junhong Zhang [1,2,*], Jun Wang [1,2,*], Xiangde Meng [2] and Dan Wang [2]**

1   State Key Laboratory of Engine, Tianjin University, Tianjin 300072, China; yuyangyang@tju.edu.cn (Y.Y.); liulehai0213@163.com (L.L.)
2   Renai College, Tianjin University, Tianjin 301636, China; Mengxiangde@126.com (X.M.); wd919006981@163.com (D.W.)
*   Correspondence: zhangjh@tju.edu.cn (J.Z.); wjun@tju.edu.cn (J.W.); Tel.: +86-136-0205-7962 (J.Z.); +86-182-2233-3117 (J.W.)

**Featured Application: This study provides a new insight for the effective prevention of explosion accidents with propane and for the development of explosion-suppression products.**

**Abstract:** The explosion-suppression effects of NSSs on overpressures, flame propagation and flame tip velocities were explored under the initial pressures of 0.2 MPa, 0.3 MPa and 0.4 MPa. All experiments tested in a constant volume combustion bomb (CVCB). Explosion reaction of premixed propane–air gas in a new designed CVCB filled with nonmetallic spherical spacers (NSSs) was analyzed. The results showed that overpressures decreased under the different initial pressures. With the increase of filling density, the overpressure decreased, the time to reach explosion overpressure decreased, and the decay rate of explosion overpressure increased. It was also found that the explosion-suppression effects of NSSs on pressures. Flame front could be captured by high-speed schlieren photography. Combustion phenomena were captured including flame propagation, corrugated laminar flame, jet flame, corrugated turbulent flame as well as tulip flame under different initial pressures. Flame tip velocities also were captured. The results demonstrate that flame tip velocities decreased with the increase of filling densities. However, compared with unfilled CVCB, flame tip velocities increased after filling NSSs in CVCB under different initial pressures. NSSs suppressed the explosion overpressure in the cylinder, and promoted the flame propagation. In both cases, NSSs played a dual role. The suppression effect of NSSs was affected by both its suppression and promotion effect on the explosion. This work provides a scientific basis for the effective prevention of explosion accidents with propane–air premixtures and the development of explosion-suppression products.

**Keywords:** explosion-suppression; nonmetallic spherical spacers; constant volume combustion bomb; overpressure; flame propagation

## 1. Introduction

Propane is a kind of high-quality energy, and has become vital alternative energy sources. In recent years, propane has been already widely used in many applications [1–3]. It is extensively used as industrial fuel. Nonetheless, combustible propane gas is highly flammable and explosive, can easily cause accidents when wrongly handled or managed during production, application, transportation and disposal. Propane explosions in confined spaces also frequently occurred, which result in facilities destruction, ecological environment and human disaster. In gas explosions, the unsteady coupling of the propagating flame and the flow field induced by the presence of blockages along the flame path produces vortices of different scales ahead of the flame front [4]. The resulting flame-vortex interaction leads to flame acceleration [5]. The unburned gas is compressed by the moving front and its pressure and temperature increase sharply. Under proper con-

ditions, uncontrollable autoignition may also occur causing deflagration-to-detonation transition (DDT).

The different type structures of suppression-explosion materials including mesh aluminum alloys (MAAs), foam materials (FMs) and nonmetallic spherical spacers (NSSs), have been used in many applications to suppress explosion in confined space. Explosion-suppression mechanism of MAAs and FMs has been extensively investigated on explosion overpressure $P_{max}$ and rate of pressure rise $dP/dt$ by many researchers [6–13]. Especially, Song [14] and Lei [15] analyzed explosion-suppression mechanism of MAAs on heat loss, and deduced the dependence between initial parameters and suppression. Joo [16] explained the quenching phenomenon of ceramic alumina foam on thermal effects, flame stretch and preferential diffusion. NSSs is another typical explosion-suppression technology that attenuates combustible gases explosion, and the better explosion-suppression performance was further confirmed than MAAs and FMs on explosion overpressure in a long cylindrical tube by Lei [15]. Lu [17] verified the explosion-suppression performance of NSSs by shock tube test, equivalent static explosion test and deflagration bomb test. Zhao [18] set up mesoscale circular tube confined space experimental bench, and confirmed that NSSs reduced the overpressure of gasoline-air mixture explosion, weakened the turbulence development and oscillating strengthening process. Flame intensity was tested in the closed tube. Although flame propagation was not completely prevented, NSSs shortened the flame duration, and reduced the flame intensity. However, most previous studies only focused on suppression of overpressure and quenching behaviors of explosion in circular tube, but NSSs are also usually filled in confined spaces such as oil tanks or gas storage. Explosion-suppression mechanism and performance of NSSs on overpressure in the confined space has not been investigated in detail, especially, flame propagation phenomenon, time history of pressure and flame velocity has not been yet determined in confined space, and it need to be further examined in depth on explosion-suppression mechanism of NSSs.

Song [14] not only verified explosion-suppression performance of NSSs on propane–air pre-mixtures with different filling densities, but also discovered that the overpressure of hydrogen explosion under the influence of NSSs increased gradually, NSSs promoted hydrogen explosion, and the dual effects of NSSs also were verified. It was generally believed that flame burning rate was enhanced when flame passed through obstacles [19–23]. The flame-obstacle interaction will result in higher flame velocity, which makes damage of combustion more complex [24]. Bychkov [25,26] developed acceleration mechanism induced by obstacles. The delayed burning between the obstacles induced flame acceleration, and jet flame in obstructed channel was produced. However, Ciccarelli [27,28] reported that the size of the hole-plates had slight effect on the flame acceleration when the hole-plate blockage ratio was low, and indicated explosion propagation in a porous medium was governed by the geometric characteristics of the porous media. The dual role between effect of NSSs on flame velocity and explosion-suppression of NSSs on overpressure has not been yet investigated in detail in a confined space. Considering the practical application of NSSs in gas storage, it is worth to investigate the double functions of NSSs on the explosion of gas mixtures.

In order to study the effect of NSSs on the propane combustion process in confined space, the explosion experiments of propane–air pre-mixtures with different initial pressures were conducted in a newly designed constant volume combustion bomb (CVCB) with different filling densities of NSSs. Current work demonstrates the following new contributions: (1) The explosion suppression mechanism of NSSs was explained and the time history of in-cylinder pressures was analyzed in confined space. It was found that NSSs promoted pressure rise at the initial propane–air premixed combustion and inhibited the pressure rise in the later reaction period. (2) NSSs with different filling densities were filled in confined space. The explosion overpressure and the explosion time were derived from the pressure-time evolution, which was used to analyze the decay rates of the propane explosion overpressures $Rp_{max}$, the rates of average pressure rise $\Phi$ and the time

$t_{max}$ to reach explosion overpressure and thus investigated the effects of NSSs on explosion-suppression. (3) Different flame propagation phenomena were all captured by high-speed Schlieren system including flame propagation, corrugated laminar flame, jet flame, corrugated turbulent flame as well as tulip flame under different initial pressures (0.2 MPa, 0.3 MPa and 0.4 MPa). Flame tip velocities were measured by the flame propagation image, and effect of NSSs with different filling densities (21.9 kg/m$^3$, 38.7 kg/m$^3$ and 45.1 kg/m$^3$) on flame propagation was analyzed at different initial pressures of 0.2~0.4 MPa. The results show that NSSs with filling density of 45.1 kg/m$^3$ had the least disturbance effect on flame propagation. (4) NSSs suppressed the explosion overpressure in the cylinder, and promoted the flame propagation. In both cases, the NSSs played a dual role. The sup-pression effect of NSSs was affected by both its suppression and promotion effect on the explosion. The results of this work might provide a new insight for the effective prevention of explosion accidents with propane and the development of explosion-suppression products.

## 2. Experimental Setup

### 2.1. Nonmetallic Spherical Spacer

The explosion-suppression materials should have the high surface area to volume ratio, it means the heat dissipation area must be large, so it has certain suppression effect on the reaction. The mutual connected tiny aperture structure can increase the probability the free radical hit the tube wall and became damaged, therefore the flame cannot carry on spreading. The unique aperture structure can absorb energy through vibration and interferes mechanically and so on to attenuate wave front pressure. According to quenching theory, the volume of the explosion suppression material should be small, the number of semicircle spacers should be large. However, for the consideration of the volume restriction in finite space, the spacers will be small and thin, which might lead to the material is not able to cope with the shock wave. Thus, NSSs requires sufficient strength. For consideration above, a thin wall skeleton structure with 28 mm diameter and eight 0.36 mm thick spacers is designed, as shown in Figure 1. Entity volume of NSSs accounts for 3.28% of space.

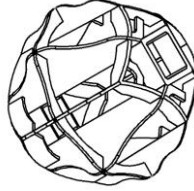

(**a**) physical mode                    (**b**) actual product

**Figure 1.** Physical mode and actual product of NSSs.

Nylon 6 has fairly high injection performance, and therefore it is selected as the substrate of NSSs. Carbon of 3–4% has been added, to ensure heat could easily conduct and transfer through NSSs. Adding 5–6% phosphor to ensure its sufficient flame retardant performance. Adding small amount of antioxidant, lubricant and plasticizer, to ensure that the composite material have good fluidity. The composite materials manufactured was tested for the corresponding material properties according to the standard, as shown in Table 1.

**Table 1.** Properties of NSSs.

| Test Item | Test Standard | Results |
|---|---|---|
| Compressive strength/MPa | ISO 527 | 135.5 |
| Tensile modulus/GPa | ISO 527 | 10.224 |
| Percentage of breaking elongation /% | ISO 527 | 1.48 |
| Melt flow index/g·(10 min)$^{-1}$ (260 °C, 2.16 kg) | ISO 1133 | 11.52 |
| Density/g·cm$^{-3}$ | ISO 1183 | 1.282 |
| Shrinkage ratio/% | ASTM D 995 | 0.6–0.8 |
| Shore hardness/HD | ISO 868 | 85 |
| Volume resistivity/$\Omega$·cm | ASTM D257 | $6.57 \times 10^{15}$ |
| Thermal conductivity/w·m$^{-1}$·K$^{-1}$ | ISO 22007 | 0.39 |

## 2.2. Experimental Apparatus

The present experiments were carried out in an improved CVCB device, and schematic was shown in Figure 2. Experimental setup equipped a highspeed Schlieren photography system, a time synchronization system, an ignition system, an image acquisition system, a pressure data acquisition system, a heating system and a gas exchange system as well as NSSs.

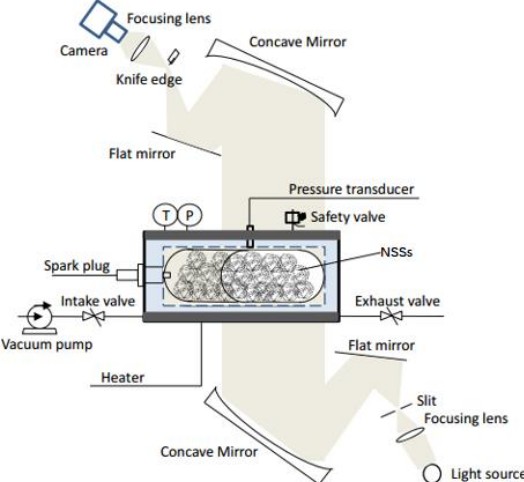

**Figure 2.** Schematic of experiment setup.

The combustion bomb of CVCB is a cylindrical cavity. Its inner diameter is 100 mm, its length is 230 mm and its volume is 2.32 L, which is similar to gas storage but with smaller size. In order to allow the light rays to penetrate front-back of CVCB, both racetrack shaped optical quartz glasses are set on both sides as optical windows. Their length are 230 mm and width are 80 mm. However, their thickness is different, the front glass is 100 mm and the back glass is 50 mm. The shadow region in Figure 2 is an optical region, which is 160-mm long and 80-mm wide. The combustion chamber is heated by electrical heating elements with total power of 2 kW. The entire vessel is uniformly heated to the specified temperature. Its temperature is kept constant with uncertainty of 4 K by a closed-loop feedback controller. The CVCB can sustain about maximum pressure of 10 MPa. An 8 MPa pressure relief valve is installed in the CVCB to keep safety. The spark plug (Bosch R6, Shanghai, China) is mounted on the left end wall to ignite mixtures and ignited duration is 0.7 ms. In order to record the pressure of in-cylinder, the pressure transducer (100 kHz, Kistler 6113 B, Shanghai, China) is arranged on the top of CVCB. The distance from the pressure transducer to the right wall is 130 mm. The in-cylinder pressure is acquired by pressure data acquisition system with the uncertainty of 0.005 MPa. The exhaust gas is scavenged by the gas exchange system. The flame images are captured and recorded by high-speed Schlieren photography system to observe the process of flame acceleration and

propagation. The ignition system, pressure data acquisition system and image acquisition system are simultaneously triggered by the time synchronizing system.

### 2.3. Experimental Parameters and Procedures

The filling density of NSSs and the initial pressure in CVCB are the important factors affecting the dual effect of NSSs on the propane–air explosion in confined space. As shown in Figure 3, three filling methods were selected: (1) The NSSs with the filling density of 21.9 kg/m$^3$ are selected for the experiment as a small amount of NSSs in CVCB (shown in Figure 3a). (2) The NSSs with the filling density of 38.7 kg/m$^3$ are selected for the experiment as a moderate amount of NSSs in CVCB (shown in Figure 3b). (3) The NSSs with the filling density of 45.1 kg/m$^3$ are selected for the experiment, that is, the NSSs are fully filled inside the CVCB (shown in Figure 3c). Therefore, the effects of three different filling densities (21.9 kg/m$^3$, 38.7 kg/m$^3$ and 45.1 kg/m$^3$) on the suppression of propane explosion were investigated in this study. Based on the Chinese standard AQ3001–2005, calculation method of the filling density is shown in Equation (1):

$$\rho = \frac{m_0}{V} \times N \tag{1}$$

where $m_0$ is the mass of a single NSSs. $N$ is the number of NSSs filled. $V$ is volume of confined space.

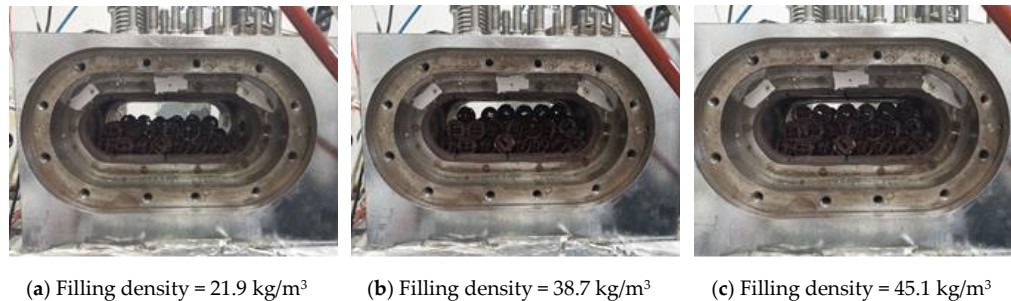

(**a**) Filling density = 21.9 kg/m$^3$     (**b**) Filling density = 38.7 kg/m$^3$     (**c**) Filling density = 45.1 kg/m$^3$

**Figure 3.** Arrangement modes of NSS with different filling densities in CVCB.

After a gas storage tank leaks, combustible gas with different volume content were formed near the leakage area, and explosions are likely to occur [29]. According to the inquiry [30], the internal pressure of the gas storage tank is 0.2~0.4 MPa when it leaks. The bomb is heated evenly to temperature 343 K. The combustion products from condensing into droplets was prevented. Experimental parameters are shown in Table 2.

**Table 2.** Experimental initial parameters.

| Initial Condition | Value |
|---|---|
| Initial temperature $T_0$ /K | $343 \pm 2$ |
| Initial pressure $P_0$ /Mpa | 0.2, 0.3, 0.4 |
| Filling density $\rho$/kg/m$^3$ | 21.9, 38.7, 45.1 |
| The equivalence ratio | 1.5 |

Before the experiment, the NSSs are placed in the combustion bomb with tweezers in order of filling from left to right and from bottom to top. Figure 3 shows the arrangement modes of NSS with different filling densities in CVCB. To reduce the accidental error from various arrangements of NSSs caused by duplicated filling, NSSs are reinstall after completing the different initial pressure experiments at same NSSs filling density. Under different initial conditions, each group of the experiment is repeated at least three times to confirm the reproducibility. Test error of each group is within 5% to ensure the accuracy of test data. Initially, the combustion chamber of CVCB is heated by the heating system to

specified temperature of 343 K. Stoichiometric propane/air mixture is obtained according to Dalton's partial pressure law. The propane/air mixture is premixed for 5 min t to make the mixture homogeneous. To avoid influence of residual gases, the combustion chamber is flushed by air at least two times after a test.

## 3. Results and Discussion

### 3.1. Variation of Pressures under the Influence of NSSs

In the previous studies [31,32], the overpressures in different devices had been tested to investigate the effects of explosion suppression materials on explosion-suppression. The explosion overpressure $P_{max}$ and the rate of pressure rise $dP/dt$ are the two important explosion parameters, which are significant for damage of gaseous explosion. In current work, the pressure transducer is used to record in-cylinder pressure in CVCB. Figure 4 illustrates the time history of in-cylinder pressures under different initial conditions, namely three different filling densities (21.9 kg/m$^3$, 38.7 kg/m$^3$ and 45.1 kg/m$^3$) and three different initial pressures (0.2 MPa, 0.3 MPa and 0.4 MPa). The in-cylinder pressure evolution curves are slightly unsmoothed due to existing inherent oscillations. Their sources are combustion noise, mechanical vibration and so on [33]. It is obvious that the pressures change with the same trend in Figure 4. Compared with unfilled CVCB, overpressures are reduced after filling NSSs in CVCB under different initial pressures. It could be explained that the microstructure of NSSs impeded explosion pressure propagation [28,34]. In CVCB, flame waves pass through surfaces of NSSs, a large amount of energy is absorbed quickly, then the reaction heat is reduced. As illustrated in literature [14], NSSs suppressed explosion reaction, retarded effective energy release. As shown in Figure 4 and Table 3, it is found that overpressures decrease with increasing filling densities of NSSs under different initial pressures, meanwhile the decay rates of overpressures $Rp_{max}$ increase with increasing filling densities of NSSs. It also indicates that the suppressing performance increases with increasing the filling density. It can be explained that number of NSSs increase, the contact surfaces are enhanced between the flame and NSSs, thus heat loss increase. It can be explained by Birk [35] that, according to the law of momentum conservation, flame passing through the explosion suppression materials, would cause internal loss and viscosity loss (Darcy).

**Table 3.** Experimental results.

| $P_0$/Mpa | $\rho$/kg/m$^3$ | $t_e$/ms | $P_e$/MPa | $\Phi_{0\_e}$/MPa/s | $t_{max}$/ms | $P_{max}$/Mpa | $\Phi_{e\_max}$/MPa/s | $Rp_{max}$/% |
|---|---|---|---|---|---|---|---|---|
| | 0 | | | 6.08 | 105 | 0.838 | 6.08 | |
| 0.2 | 21.9 | 24 | 0.576 | 15.67 | 54.5 | 0.745 | 5.54 | 11.10 |
| | 38.7 | 11.5 | 0.521 | 27.91 | 42.5 | 0.661 | 4.52 | 21.12 |
| | 45.1 | 6.5 | 0.531 | 50.92 | 30.5 | 0.602 | 2.96 | 28.16 |
| | 0 | | | 12.91 | 79.5 | 1.326 | 12.91 | |
| 0.3 | 21.9 | 17 | 1.005 | 41.47 | 38.5 | 1.173 | 7.81 | 11.54 |
| | 38.7 | 14 | 0.846 | 39 | 33 | 0.943 | 5.11 | 28.88 |
| | 45.1 | 14 | 0.813 | 36.64 | 31.5 | 0.875 | 3.54 | 34.01 |
| | 0 | | | 15.12 | 92 | 1.791 | 15.12 | |
| 0.4 | 21.9 | 14.5 | 1.067 | 46 | 61.5 | 1.467 | 8.51 | 18.10 |
| | 38.7 | 10.5 | 1.007 | 57.81 | 30.5 | 1.15 | 7.15 | 35.80 |
| | 45.1 | 7.5 | 0.871 | 62.8 | 27.5 | 1.008 | 6.85 | 43.72 |

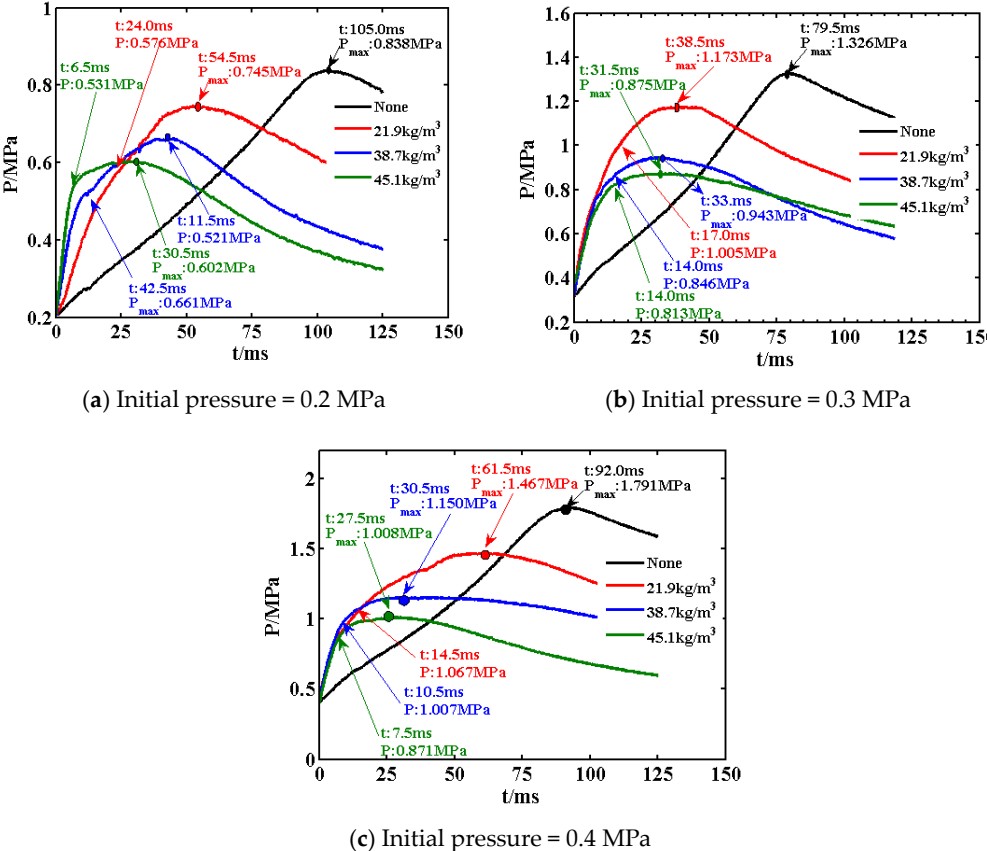

(**a**) Initial pressure = 0.2 MPa    (**b**) Initial pressure = 0.3 MPa

(**c**) Initial pressure = 0.4 MPa

**Figure 4.** Time history of in-cylinder pressures at initial pressures of 0.2 MPa, 0.3 MPa and 0.4 MPa.

The decay rates of explosion overpressures $Rp_{max}$ and the rates of average pressure rise $\Phi$ are analyzed to illustrate the effects of NSSs on explosion-suppression, the calculation formulas of $Rp_{max}$ and $\Phi$ are expressed as follows.

$$Rp_{max} = \frac{P_{max}^0 - P_{max}^1}{P_{max}^0} \times 100\%,$$  (2)

$$\begin{aligned} \Phi &= \frac{\Delta P}{\Delta t} \\ \Phi_{0\_e} &= \frac{\Delta P}{\Delta t} = \frac{P_e - P_0}{t_e - 0} \\ \Phi_{e\_max} &= \frac{\Delta P}{\Delta t} = \frac{P_{max} - P_e}{t_{max} - t_e} \end{aligned}$$  (3)

where $P_{max}^1$ is overpressure in the filled CVCB, $P_{max}^0$ is overpressure in the unfilled CVCB, $\Phi_{0\_e}$ is the rate of average pressure rise when propane and air occurred reaction early, the time is 0 to $t_e$. $\Phi_{e\_max}$ is the rate of average pressure rise when NSSs retarded effective energy release, the time is $t_e$ to $t_{max}$. $P_e$ is pressure when the time is $t_e$.

Corresponding overpressure $P_{max}$ of in-cylinder and some important features from Figure 4 are summarized in Table 3. When the filling density is 21.9 kg/m$^3$ and the initial pressures are 0.2 MPa, 0.3 MPa and 0.4 MPa, the corresponding decay rates of overpressures $Rp_{max}$ are 11.10%, 11.54% and 18.10%, respectively. When the filling density is 38.7 kg/m$^3$ and the initial pressures are 0.2 MPa, 0.3 MPa and 0.4 MPa, the corresponding decay rates of overpressures $Rp_{max}$ are 21.12%, 28.88% and 35.80%, respectively. When the filling density is 45.1 kg/m$^3$ and the initial pressures are 0.2 MPa, 0.3 MPa and 0.4 MPa, the corresponding decay rates of overpressures $Rp_{max}$ are 28.16%, 34.01% and 43.72%, respectively. Under different filling densities, the decay rates of overpressures $Rp_{max}$ increase with increasing initial pressures which is an interesting phenomenon that deserves further investigation. It is obvious that the time histories of in-cylinder pressures in the unfilled CVCB are approximately straight line, there is no obvious inflection point, thus we believe that $\Phi_{0\_e}$ is approximately equal to $\Phi_{e\_max}$. The time histories of in-cylinder

pressures in the filled CVCB have an inflection point, $t_e$ is the moment corresponding to the inflection point $P_e$. From 0 to $t_e$, propane and air occur reaction early, the beginning of the flame propagation process, flame propagation has less contact surface with NSSs. Rates of average pressures rise $\Phi_{0\_e}$ are much greater than that without filling due to the presence of obstacles. A possible explanation for this might be that NSSs are playing roles as multiple obstacles. These obstacles promote the formation of turbulence, resulted in flame acceleration, and induce earlier rise of dP/dt [36]. NSSs acts as a turbulence generator, which plays positive roles in affecting the explosion [37]. As flame continues to spread from $t_e$ to $t_{max}$, NSSs retard effective energy release. Rates of average pressures rise $\Phi_{0\_e}$ are reduced rapidly. Compared with unfilled CVCB, rates of average pressures rise $\Phi_{0\_e}$ become smaller after filling NSSs in CVCB. As described by Zhuang [38] and Song [14], a very high surface efficiency is formed in a unit volume by structure of NSSs with excellent thermal conductivity, and absorbed a good deal of heat. Thus, it could form significantly reduce reaction rate. At the same time, the time $t_{max}$ to reach explosion overpressure is smaller than that with unfilled CVCB, and overpressures appear earlier than that with unfilled CVCB. A possible explanation for this might be that the reaction has been effectively reduced by NSSs including increasing residual reactant. The skeleton structure suppresses propane from cracking into more radicals, and thus leads to a higher amount of residual propane.

To sum up, NSSs play both positive and negative roles in affecting the explosion. On one hand, when flame propagation has less contact surface with NSSs, they act as turbulence generator and facilitate explosion process, causing the rate of pressure rise increase. The skeleton structure of NSSs shows a positive effect on the explosion. On the other hand, as the flame propagation, the contact surface between NSSs and the flame increase. The skeleton structure of NSSs can form a very high surface efficiency and achieve significant heat dissipation, causing overpressure decreases.

### 3.2. Effect of NSSs on Explosion Flame Propagation

Generally, the flame structure and propagation will be affected by obstacles in a confined space. In our previous work [39], the explosion experiments of propane–air pre-mixtures were conducted in cylinders with different filling densities of NSSs (21.9 kg/m³, 38.7 kg/m³ and 45.1 kg/m³). The effect of filling densities of NSSs on flame propagation characteristics and turbulence in CVCB with different equivalence ratios of propane–air were analyzed in detail. The results show that NSSs had promotion effect on the flame propagation process. There was no mushroom-like flame formed in the cylinder under each equivalent ratio when the filling density of NSSs was the maximum (45.1 kg/m³). The flame propagation process was the most stable, NSSs had the least perturbation effect on the flame. With the increase of filling density, the velocity oscillation degree decreased. In the present work, as shown in Figure 2, the position and time of flame just entering the shadow region (observation window) at the X axis direction represent the zero point. The right end wall of the CVCB is located 150 mm from the zero point. Figure 5 illustrates a series of high-speed photographs of flame propagation for the propane explosion in CVCB with NSSs (21.9 kg/m³) under different initial pressures (0.2 MPa, 0.3 MPa and 0.4 MPa). The flame propagated from the left to the right.

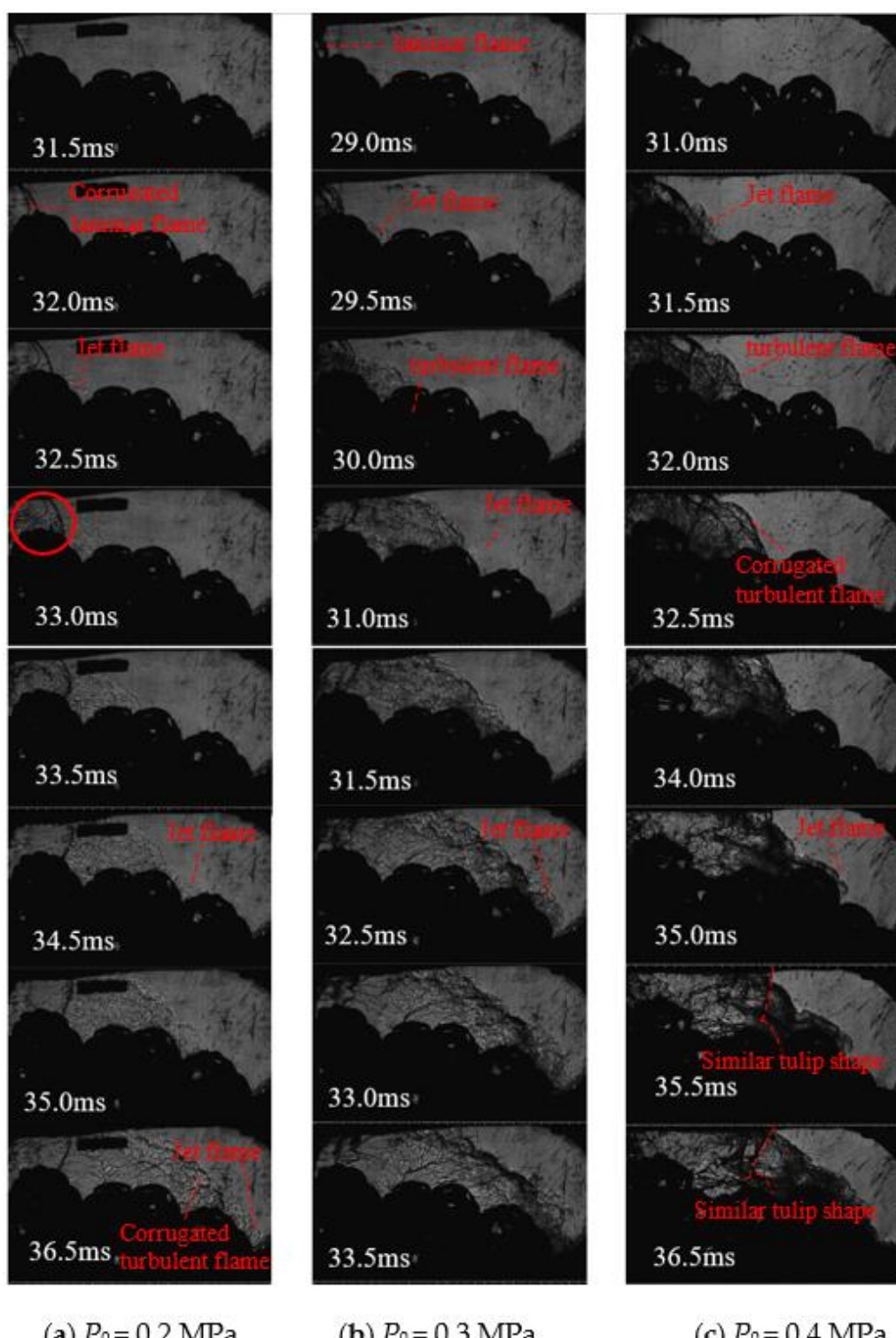

(a) $P_0 = 0.2$ MPa     (b) $P_0 = 0.3$ MPa     (c) $P_0 = 0.4$ MPa

**Figure 5.** The process of flame propagation under three different initial pressures when the filling density is 21.9 kg/m$^3$.

In Figure 5, it is obvious that the flame structure changes with the similar trend under different initial pressures. It can be seen that the development of the flame in the present work was divided into three obvious structures: laminar flame propagation, jet flame formation and turbulent flame development. As the flame develops, the structures of the flame are slightly different under different initial pressures. When initial pressure is 0.2 MPa, at 31.5 ms and 32.0 ms, areas not filled with NSSs show corrugated laminar flame

within the observation window. At 32.5 ms, flame begins to fold at the junction of unfilled area and NSSs, and the disturbance region is formed due to jet flow. Thus, the flame rapidly turns into turbulent flame and propagates forward. At 33.0 ms, two type flames just start to merge at the place marked by the circle. A possible explanation for this might be that flame and vortex interacted together, and consequent flame folding, and wrinkling were the main mechanism for the increase in the flame surface area [40]. At 34.5 ms, due to the disturbance of NSSs, it shows that the vortex performs a half-mushroom-like front around the jet flame. At 36.5 ms, the flame spreads to the end region of the CVCB and shows corrugated turbulent flame and jet turbulent flame. As shown in Figure 6a, when the jet turbulent flame is formed, the flame tip velocity begins increasing again in the position of approximate 130 mm. When initial pressures are 0.3 MPa and 0.4 MPa, jet turbulent flames also appear at the position, as the initial pressure of 0.2 MPa. This is the reason for placement of the NSSs. Compared with 0.2 MPa, turbulences are stronger and there are fewer areas of laminar within the observation window flow at 0.3 MPa and 0.4 MPa. When initial pressure is 0.4 MPa, at 32.5 ms, corrugated turbulent flame is formed. At 35.5 ms and 36.5 ms, similar tulip shapes are formed. It is verified that the turbulence intensity at 0.4 MPa is greater than 0.2 MPa and 0.3 MPa. When the filling density is 21.9 kg/m$^3$, flame propagation after a small amount of NSSs leaded to produce turbulent vortex with a larger eddy structure. This is also consistent with the situation in literature [41]. The interconnectivity of NSSs can provide many narrow passageways, and the flame is divided into a good deal of flame streams immediately because of skeleton structures of NSSs. It also further verified the viewpoint of Wang [37] that obstacles played positive roles in affecting the explosion. To sum up, the turbulence is generated when flame pass through narrow passageways of NSSs, the turbulence intensity increases with increasing initial pressure in filled CVCB, and more intense flame phenomenon is induced with increasing initial pressure. It is further verified that promoting effect of NSSs on flame propagation.

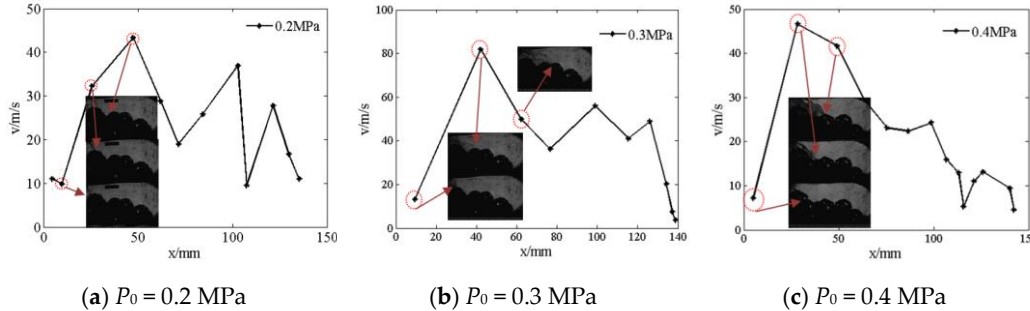

(**a**) $P_0$ = 0.2 MPa　　　　　　(**b**) $P_0$ = 0.3 MPa　　　　　　(**c**) $P_0$ = 0.4 MPa

**Figure 6.** Flame tip velocity at initial pressures 0.2 MPa, 0.3 MPa and 0.4 MPa.

### 3.3. Effect of Initial Pressure on Flame Tip Velocity

Flame tip velocity was defined, which had detailed discussion in previous work [42]. Flame tip velocity is obtained by the difference of flame front position between consecutive images. In the current work, 8000 images per second are collected by the image acquisition devices, namely the time internal of consecutive images is 0.125 ms. The uncertainty is no more than 0.18 mm on flame front location. As a result, the uncertainty is no more than 1.44 m/s on flame velocity. The flame tip velocities are obtained at 21.9 kg/m$^3$ filling density under different initial pressures in Figure 6.

As shown in Figure 6, at the same filling density (21.9 kg/m$^3$), the flame tip velocity in the cylinder filled with NSSs area is greater than that in the unfilled NSSs area. The degree of the burned gas expansion in the lower part of the cylinder filled with NSSs are greater than that in the upper part of the cylinder without filling NSSs. This is due to the flame was affected by skeleton structures of NSSs, the limitation of space and the expansion of burned products. In addition, due to the faster expansion of the burned products, as show in Figure 6, at the top position of the curve, the jet flame gains a faster velocity of 43.34 m/s (47.51 mm), 81.88 m/s (42.16 mm) and 46.64 m/s (28.44 mm) under different

initial pressures (0.2 MPa, 0.3 MPa and 0.4 MPa). The velocity variation under different initial pressures was consistent with the flame passing through hole size of 2 mm and porosity of 12% orifice in doctoral thesis [43]. On the basis of the Bychkov theory [26], KH/RT instability with shear effect were triggered that played a vital role on increasing flame surface area and flame burning velocity. It is also further verified that flame velocity increases in the filled with NSSs area, and NSSs promote flame propagation.

### 3.4. Effect of Filling Density on Flame Tip Velocity

The average flame tip velocities at different initial pressures and different filling densities in CVCB are obtained, as show in Table 4. The average flame tip velocities filled with NSSs are greater than that of unfilled NSSs in CVCB. The average flame tip velocity reaches maximum when the filling density was 21.9 kg/m$^3$, and initial pressure has little effect on average of flame tip velocity. The average flame tip velocities of 38.7 kg/m$^3$ and 45.1 kg/m$^3$ are significantly lower than that of 21.9 kg/m$^3$. With the increase of initial pressure, the average flame tip velocity decreases. This is positively correlated with the decay rates of the propane explosion overpressures which was increasing as the initial pressure increased. The average flame tip velocity is maximum when the filling density is 21.9 kg/m$^3$, a possible explanation for this might be that less NSSs are stacked together to form the larger gaps and spaces, which enhances the turbulence of flame combustion. During the process, as Korzhavin' work [44], the mixture would be involved in the eddy turbulence, and the flame had a higher velocity.

**Table 4.** Average flame tip velocities at different filling densities.

| $\rho$/kg/m$^3$ | $P_0$/MPa | v/m/s |
|---|---|---|
| 0 | 0.2 | 2.17 |
|  | 0.3 | 0.81 |
|  | 0.4 | 1.85 |
| 21.9 | 0.2 | 22.38 |
|  | 0.3 | 25.84 |
|  | 0.4 | 22.56 |
| 38.7 | 0.2 | 14.49 |
|  | 0.3 | 9.11 |
|  | 0.4 | 6.73 |
| 45.1 | 0.2 | 15.34 |
|  | 0.3 | 12.56 |
|  | 0.4 | 11.49 |

Compared with the filling density of 21.9 kg/m$^3$, the average flame tip velocities of 38.1 kg/m$^3$ and 45.1 kg/m$^3$ decrease. It is well explained that when the filling density of NSSs are large, there is less space for the flame vortex to be broken and the flame front to stretch, and the flame existed behind the skeleton structures of NSSs, led to the flame velocity decrease. Similar to Oh's [45] study, the flame velocity would decrease just behind the plate obstacle because of the obstacle-induced eddy momentum.

In Figure 7, NSSs have the same influence on flame propagation, so after filling NSSs, the flame tip velocities have different oscillation degree in the cylinder. When the filling density is 21.9 kg/m$^3$, the maximum flame tip velocity is larger and the oscillation degree of the flame tip velocity is greater than that when the filling density are 38.7 kg/m$^3$ and 45.1 kg/m$^3$. This was related to the distribution state of NSSs. Thus, it is also verified that the average flame tip velocities because of filling NSSs are greater than that of unfilled NSSs in CVCB. However, the average flame tip velocity decreases with increasing filling density of NSSs.

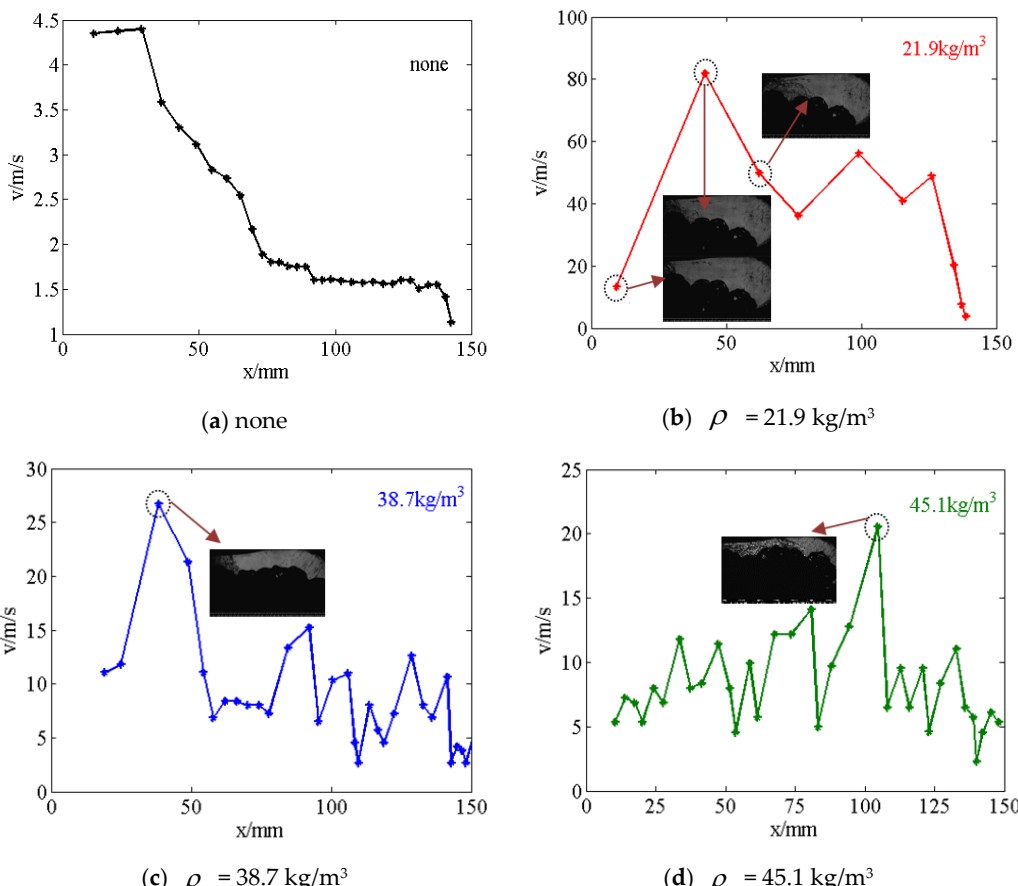

**Figure 7.** Flame tip velocity under different filling densities at initial pressure of 0.3 MPa.

To sum up, by combining overpressure and flame velocity, NSSs had promotion/suppressing double functions on the explosion of propane–air mixtures. NSSs showed a better suppressing effect on overpressure, but NSSs had a positive effect on the flame propagation. Thus, the suppression effect of NSSs was influenced by its suppression and promotion effect on explosion of propane–air mixtures. With the increase of filling density, the stronger the suppressing effect of overpressure was, the weaker the promoting effect on the velocity was, and the better the comprehensive suppressing effect was. The results of this work might provide a new insight for the explosion-suppression mechanism of NSSs in confined space such as gas tank, and provide a new scientific basis for application of NSSs explosion-suppression.

## 4. Conclusions

A newly designed constant volume combustion bomb (CVCB) equipped was employed in the present work to investigate the suppression effects of NSSs on propane–air explosion. Combustion phenomena were all captured by high-speed Schlieren photography including flame propagation. In addition, explosion overpressure, flame tip velocities and the average flame tip velocities at different filling densities and initial pressures was obtained. The main conclusions drawn from this work are described as follows.

(1)    It was found that NSSs promoted pressure rise in the initial propane–air premixed combustion and inhibited the pressure rise in the later reaction period. Thus, the result of overpressure decreased in confined space due to heat loss was caused by huge amounts of NSSs. NSSs also suppress propane from cracking into more radicals and thus led to a higher amount of residual propane. Thus, it was explained that the explosion-suppression effects of NSSs on pressures and the explosion suppression mechanism of NSSs. Under different initial pressures (0.2 MPa, 0.3 MPa and 0.4 MPa),

with the increase of filling density, the overpressure decreased, the time to reach explosion overpressure decreased and the decay rate of explosion overpressure $Rp_{max}$ increased. For this experimental environment, NSSs had the best explosion suppression effect when the filling density was 45.1 kg/m$^3$ under different initial pressures.

(2)　Under different initial pressures, flame passed through NSSs, the flame expanded and accelerated, and the NSSs acted as obstacles. When the filling density of NSSs was 21.9 kg/m$^3$, the unfilled area was large. Further, the flame passed through the NSSs, which formed a vortex. The breakage of the vortex and the stretching of the flame array made the flame accelerate faster. When the filling densities of the NSSs were 38.7 kg/m$^3$ and 45.1 kg/m$^3$, the unfilled area was small, so that most of the flame exists in the rear of the NSSs, leading to the decrease of the flame velocity, and the oscillation degree of the flame tip velocities decreased. This was positively correlated with the increase of the decay rate of explosion overpressures. It was verified that NSSs promoted effect on flame velocity and suppressed effect of filling density on flame velocity.

(3)　On one hand, NSSs suppressed the explosion overpressure in the cylinder, the higher density of NSSs was, the stronger their suppression effect was; however, on the other hand, NSSs promoted the flame propagation. In both cases, the NSSs played a dual role on explosion suppression or promotion. However, it is important to illustrated that its double role can also be available for the experimental set-u, similar to equipment used in this investigation.

(4)　NSSs are also usually filled in gas storage with two perforated plates to prevent flammable gas explosion. The flame passes through the perforated plates, and accelerates, thus different combustion models including local autoignition and end gas autoignition will be induced by compression effect. As a result, the explosion-suppression mechanism of NSSs at the condition of flame acceleration in confined spaces with perforated plates can be further investigated in detail. The explosion-suppression effects of NSSs on different combustion models should be further investigated in depth to prevent flammable gas explosion under different combustion environments or intensive combustion environments.

**Author Contributions:** Y.Y. developed model of nonmetallic spherical spacers (NSSs), conducted the analyses and wrote the paper; L.L. did experiments on the NSSs; J.Z. contributed analysis tools; X.M. developed the NSSs actual product; J.W. and D.W. revised the paper. All authors have read and agreed to the published version of the manuscript.

**Funding:** The financial support from Tianjin Science and Technology Program Project (20YDT-PJC02020) and Scientific and Technological Research Program of TianJin Municipal Education Commission (2019KJ152).

**Acknowledgments:** The authors wish to acknowledge the financial support from Tianjin Science and Technology Program Project (20YDTPJC02020) and Scientific and Technological Research Program of TianJin Municipal Education Commission (2019KJ152). All individuals included in this section have consented to the acknowledgement.

**Conflicts of Interest:** The authors declare no conflict of interest.

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
