# Peer review of "Explosion-Suppression Characteristics of Nonmetallic Spherical Spacers on Propane-Air Mixtures in Confined Space"

_applsci, doi:10.3390/app11199238_

Round 1

Reviewer 1 Report

The paper, devoted to an actual theme of explosion prevention and suppression, describes the results of an experimental study of the effect of nonmetallic spherical spacers (NSSs) on flame propagation in the specific small closed cylindrical volume (its inner diameter was 10 cm and length 23 cm). An initial pressure inside this cylinder partially or completely filled with NSSs was 2, 3 or 4 bars. Loading density of NSSs was approximately 22, 39 and 45 kg/m3. The latter value corresponds to nearly full filling of the cylinder volume by NSSs. At smaller loading densities of 22 and 39 kg/m3 the flame could propagate above the layer of NSSs.

                The only (obvious) conclusion of the study that can be extrapolated to another geometry or situation is that the higher the loading density of NSSs is, the stronger their suppression effect is. On the other hand, when NSSs fill the cylinder volume partially, they promote flame propagation. Thus, this last conclusion is strictly specific for the considered experimental set-up and hardly can be extrapolated to another geometry or situation.

                It is worth to mention the above statement in the revised version.

                However, it is worth to publish the presented database after the correction of the English used in the considered paper.

Reviewer 2 Report

This is an interesting paper addressing an important issue. It can be considered for publication in Applied Sciences. However, I have the following comments that the authors should carefully implement in the revised manuscript before publication.

1) Introduction - These sentences

“Explosions usually occur, which is caused by flame acceleration. The unburned gas is compressed and its pressure and temperature increase sharply. Under proper conditions, autoignition occurs. Uncontrollable autoignition thus deflagration, even to detonation transition (DDT).”

should be corrected to

“In gas explosions, the unsteady coupling of the propagating flame and the flow field induced by the presence of blockages along the flame path produces vortices of different scales ahead of the flame front [x]. The resulting flame-vortex interaction leads to flame acceleration [xx]. The unburned gas is compressed by the moving front and its pressure and temperature increase sharply. Under proper conditions, uncontrollable autoignition may also occur causing deflagration-to-detonation transition (DDT).” with [x] = Journal of Hazardous Materials, Volume 180, Issue 1-3, 2010, Pages 71-78 and [xx] = Chemical Engineering Science, Volume 71, 2012, Pages 539-551.

2) Introduction - The connection between the aim of the work and the literature gaps should be more deeply discussed, thus giving more strength to the reason behind this work.

3) Figure 4 should be improved, as now it is not clear.

4) Results and Discussion/Conclusions - The practical impact of the results obtained in this work should be better highlighted.

5) Conclusions - The authors should also give an outlook on future research work.

I’m willing to review the revised manuscript.

Round 2

Reviewer 2 Report

The authors have addressed my comments in a satisfactory manner. Overall, the manuscript has been improved after revisions.

Please, before publication, double-check the references: in (at least) two cases, in the revised manuscript, surnames have been confused with forenames.